# Phage Targeting *Streptococcus mutans* In Vitro and In Vivo as a Caries-Preventive Modality

**DOI:** 10.3390/antibiotics10081015

**Published:** 2021-08-21

**Authors:** Amit Wolfoviz-Zilberman, Reut Kraitman, Ronen Hazan, Michael Friedman, Yael Houri-Haddad, Nurit Beyth

**Affiliations:** 1Department of Prosthodontics, Hadassah Medical Center, Faculty of Dental Medicine, Hebrew University of Jerusalem, Jerusalem 9112001, Israel; amit.wolfoviz@mail.huji.ac.il (A.W.-Z.); reut.kraitman@mail.huji.ac.il (R.K.); yaelho@ekmd.huji.ac.il (Y.H.-H.); 2Institute of Biomedical and Oral Research (IBOR), Faculty of Dental Medicine, Hebrew University of Jerusalem, Jerusalem 9112001, Israel; ronenh@ekmd.huji.ac.il; 3Department of Pharmaceutics, The Institute for Drug Research, Faculty of Medicine, The Hebrew University, Jerusalem 91120, Israel; michaelf@ekmd.huji.ac.il

**Keywords:** dental caries, phage therapy, *S. mutans*, bacteriophage, *S. mutans* phage

## Abstract

Dental caries is a common infectious disease worldwide. Current conventional therapies lack specific antimicrobial effects against *Streptococcus mutans*, a key bacterium that induces caries. A promising alternative approach is bacteriophage (phage) therapy. Recently, SMHBZ8 phage targeting *S. mutans* was isolated and characterized. The aim of this study was to evaluate the caries-prevention efficacy of SMHBZ8 using in vitro and in vivo caries models. Hemi-mandibles dissected from euthanized healthy mice were subjected to caries-promoting conditions in vitro. Jaws treated with phage therapy in suspension and in formulation with a sustained-release delivery system showed no carious lesions, similar to control and chlorhexidine-treated jaws. Subsequently, SMHBZ8 phage suspension also prevented carious lesion development in a murine caries model in vivo. In both models, caries lesions were analyzed clinically and radiographically by µCT scans. This study shows how SMHBZ8 phage therapy targeting *S. mutans* can serve as an efficient caries-prevention modality, in suspension or with a sustained-release delivery system, by in vitro and in vivo mouse models.

## 1. Introduction

Dental caries, known as tooth decay, is a common infectious disease worldwide. Dental caries is characterized by demineralization of enamel and dentin via acid generated by bacterial biofilm. Oral bacteria metabolize carbohydrates, producing organic acids that diffuse into the enamel and dentin and dissolve minerals [1].

Dental caries cannot be easily treated with conventional therapies, due to the formation of cariogenic bacteria biofilm, which protects the bacteria by preventing the penetration of conventional antimicrobial modalities [2,3]. One of the most widely used antiseptics, especially in dentistry, is chlorohexidine (CHX). Various studies have showed its efficacy in reducing plaque accumulation, but it also causes colorization of teeth and exhibits a cytotoxic effect on human cells [4].

Current therapies lack sensitivity; they are not species-specific and kill pathogenic species, as well as commensal species that are protective against the formation of pathogenic biofilms. Consequently, there is a need to develop therapeutic strategies that are species-specific for cariogenic pathogens.

Recent advances in antimicrobial materials research are aimed at developing new, highly effective materials that are species-specific for cariogenic bacteria, with a low risk of causing resistance.

Despite advances in science, dental caries, caused by multiple pathogenic bacterial species, is still a costly disease. A key bacterium that induces caries is *Streptococcus mutans* [5]. Therefore, inhibiting *S. mutans* growth and biofilm formation may delay the caries development process.

A promising alternative approach is bacteriophage (phage) therapy. Bacteriophages (phages) are viruses that infect bacteria by inserting their DNA, causing cell lysis. The main advantage of phage therapy is the ability to target specific bacterial strains; thus, it does not significantly impact the commensal oral flora. Moreover, phages have the ability to penetrate and disrupt biofilms, which is a key benefit when targeting cariogenic bacteria [6,7].

In the past few years, various studies have tried to isolate and characterize phages against *S. mutans* [8,9,10,11,12]; thus far, only three have been isolated and their genomes sequenced: M102 [8,10], M102AD [9], and ΦAPCM01 [10]. Recently, Hadar Ben-Zaken et al. [11] isolated and characterized a new phage targeting *S. mutans*, SMHBZ8, which effectively infects and kills both planktonic and biofilm cultures of *S. mutans* in vitro.

One concern regarding the use of phages is their stability in solution. Phages have limited stability in solution, and there can be a significant reduction in phage titer during processing and storage [13]. Due to the fact that phages are protein structures, they are susceptible to protein denaturation factors, such as high temperature, pH, and organic solvents [13]. It was suggested that a sustained-release formulation of phages may help prolong their efficacy [13,14]. González-Menéndez et al. [15], for example, recommend encapsulating phages to maintain their stability and as a suitable method for shipment.

Sustained-release dosage forms of conventional oral care modalities have been suggested, and some formulations have been used for decades, e.g., PerioChip^®^. Oral varnishes offer a further option; these forms are applied to teeth or mucosa and solidify into a film that releases drugs in a sustained manner. These varnishes contain various pharmaceutical polymers, including cellulose derivatives, such as hydroxypropyl cellulose. Varnishes with a variety of drugs, including clotrimazole, chlorhexidine, triclosan, cetylpyridinium chloride, and others, have been reported in the literature [16,17,18,19,20]. However, the potential of sustained-release phage treatment against *S. mutans* has not yet been investigated.

The objectives of this study were to examine the efficacy of SMHBZ8 phage as a caries-preventive modality by a new quantitative and reproducible in vitro caries model [21] using the jaws of euthanized healthy mice that were sacrificed for other research purposes and no longer needed (for example, control group mice). An in vivo caries model was designed to evaluate and verify the in vitro model’s results.

We evaluated a hydroxypropyl-cellulose-based varnish formulation in a sustained-release delivery system for SMHBZ8 phage. We hypothesized that SMHBZ8 phage could target *S. mutans* and prevent caries induction effectively when given in suspension or as a sustained-release drug.

## 2. Results

### 2.1. Assessment of Phage Lytic Activity against Streptococcus Mutans

As shown in Figure 1a, after 24 and 48 h, phages in suspension or by slow-release polymer induced a decrease in SMHBZ bacterial load. As expected, a similar and significant decrease (~3 log decrease) was shown in the phage and phage:polymer groups compared to the groups infected with bacteria only (SMHBZ) and treated with polymer (SMHBZ + polymer).

### 2.2. Phages and Polymer Are Not Toxic to Cells

The viability of RAW macrophage mouse cells was unaffected by treatment with the phage:polymer formulation, demonstrating similar viability rates (*p* = 5) to untreated cells (control) (Figure 2).

### 2.3. Phage Lytic Activity against S. mutans Prevented Caries Development In Vitro in Extracted-Jaw Caries Model

#### 2.3.1. Clinical Evaluation

Caries lesions were detected clinically in the group infected with SMHBZ bacteria within 5 days. Representative images of an average lesion before and after bacterial infection are shown in Figure 3a,b. No carious lesions were detected in any treatment group: phage suspension, sustained-release phage:polymer, chlorhexidine mouthwash 0.2%, and chlorhexidine dental gel 1%.

#### 2.3.2. Quantification of Demineralization

µCT scans demonstrated carious lesions in the group infected with bacteria, while teeth in all treatment groups remained intact, depending on the clinical appearance. Representative radiographic images 5 days after bacterial infection are shown in Figure 3c.

In the group infected with bacteria, radiographic images demonstrate extensive dentin lesions in fissures (Figure 3 C1), while the teeth in the treatment groups remained intact (Figure 3 C2–C5).

Figure 4 shows the differences in the caries density range after inducing caries-promoting conditions. A significant decrease was observed in all treatment groups compared to the untreated group infected with bacteria (*p* < 0.5).

#### 2.3.3. Bacterial Outgrowth Evaluation

At the endpoint of the experiments, viable bacterial count in the group infected with bacteria was around 10^6^ colony forming units (CFU)/mL, while a significant decrease was shown in all experimental groups (*p* < 0.05) (Figure 5).

#### 2.3.4. Caries Scoring Using a Novel Scoring Method over Hemi-Sectioned Molars

As shown in Figure 6, caries scores for hemi-sectioned molars increased significantly (*p* = 0.011) in the jaws infected with bacteria (*n* = 5) compared to jaws treated with phages (in suspension or with polymer) or CHX (mouthwash or dental gel) (*n* = 5).

### 2.4. Phage Lytic Activity against S. mutans Prevented Caries Development in Experimental In Vivo Caries Model

#### 2.4.1. Clinical Evaluation

Clinical photographs demonstrate carious lesion development in the group infected with bacteria, while the teeth in the groups treated with CHX and phage remained intact. Representative photographs are shown in Figure 7a.

#### 2.4.2. Quantification of Demineralization

µCT radiograph scans demonstrated extensive carious lesions in the group infected with *UA159* bacteria, while, in the control (no bacteria) and CHX- and phage-treated groups, no demineralization was shown. Representative radiographic images are shown in Figure 7b.

When comparing the total cubic volume of the first molar’s crown divided into six selected density ranges (by mgHA/ccm), a significant increase (*p* < 0.05) in the volumetric percentage of density representing carious dentin (500–1500 mgHA/ccm) was observed in the untreated group with bacterial infection (Figure 8).

Figure 7b shows µCT radiograph scans emphasizing the clinical appearance.

## 3. Discussion

Phage therapy targeting *S. mutans* was shown to be an efficient caries-prevention modality by in vitro and in vivo mouse models. Herein, we demonstrated how phage therapy against *S. mutans* showed similar efficacy in reducing bacterial load and preventing carious lesion development compared to conventional treatment using chlorhexidine mouthwash or dental gel. As shown in Figure 3 and Figure 7, phage therapy and chlorhexidine succeeded in preventing demineralization in vitro and in vivo, respectively.

Interestingly, SMHBZ8 phage was equally effective at reducing the *S. mutans* bacterial load and preventing caries when used in suspension and sustained-release formulation. The reason could be that phages are self-replicating species in the presence of target bacteria; thus, their concentration remains high as long as the infecting bacteria are present. The sustained-release formulation may be more effective in preventing *S. mutans* colonization, as the oral residence time of the suspension is significantly shorter than that of the sustained-release varnish.

Thus, the polymer–phage formulation, which shows caries-prevention efficacy with no evidence of toxicity to cells, can be administered conveniently in the dental clinic. This formulation can be applied topically on the occlusal surfaces of teeth or as a liner under restorations, which cannot be done using phages in suspension.

As mentioned before, as far as we know, only four phages against *S. mutans* have been isolated and characterized [8,9,10,11]. Ben-Zaken et al. [11] showed that SMHBZ8 phage was efficient in reducing *S. mutans* bacterial load in a dentin model, but there has been no evidence for phage therapy against *S. mutans* examining its ability to prevent caries or demineralization in vitro or in vivo. The current study is the first to evaluate the efficiency of phage therapy as a caries-prevention modality.

Antibacterial materials that target cariogenic bacteria, especially *Streptococcus mutans*, are at the center of cariology research; conventional treatments lack the ability to penetrate dental biofilm and reduce the cariogenic bacterial load without causing unwanted side effects. For example, chlorhexidine, which is widely used and investigated, exhibits bactericidal effects, but also causes teeth discoloration and is cytotoxic to human cells in high concentrations [4,22,23]. Furthermore, various conventional modalities for caries prevention are not targeted to reduce *S. mutans* bacterial load; fluoride-containing mouthwashes and toothpastes have shown biological effects on acid production and glucan synthesis in *S. mutans* in vitro, but their biological effects in vivo remain uncertain [3,24,25,26].

Antibiotics are rarely used in oral diseases, as they have a limited effect on biofilms, while phages have shown the ability to inhibit bacterial growth in biofilm [10,11,27]. Moreover, antibiotic therapy is not species-specific and can affect pathogenic, as well as commensal species. Therefore, phages, which are species-specific, may be a promising alternative approach for caries treatment and prevention [6,7,27].

Maske et al. [28] reviewed in vitro dental caries biofilm models. Mimicking the oral cavity environment in vitro is complex, thus models show high variability; a significant portion of the in vitro models described used human enamel slabs or extracted teeth. Demineralization is difficult to diagnose and evaluate using these models, and often requires unique equipment with high technique sensitivity, such as microhardness [26,29,30], atomic force microscopy [31], polarized light microscopy [30,32], and quantitative light fluorescence (QLF) [22].

Both models described in the present study enable the identification of carious lesions and quantification of demineralization by demonstrating the lesions clinically and radiographically, corresponding to the identification of carious lesions by dentists. Furthermore, these methods require only a stereomicroscope and µCT, the analysis is relatively simple, and the results are repeatable.

Our in vitro model using extracted jaws from dissected healthy mice (obtained from euthanized control animals from parallel experiments) offers a superior alternative compared to other models, not only because of its relatively rapid caries production, but also because it allowed us to control multiple criteria, such as age, genetics, growth conditions, diet, and the starting condition of the dental hard tissues. The morphology of murine molar teeth resembles the morphology of human molars, containing sulci and fissures [33]. Nevertheless, they are smaller and the dental hard tissues are thinner than those of human teeth. Hence, carious lesion development in murine teeth is enhanced compared to the process in human teeth or enamel slabs [34].

SMHBZ8 phage therapy effectively targeted *S. mutans* in vitro and in vivo, and thus succeeded in preventing caries. In vivo SMHBZ8 phage treatment using oral swabs showed significant inhibition of demineralization and carious lesion development. In future investigations, in vivo studies should aim to evaluate sustained-release SMHBZ8 in a phage:polymer formulation. It would also be interesting to examine combined treatment of SMHBZ8 phage and chlorhexidine. In this study, the phages remained viable in the presence of chlorhexidine, and we showed how phage therapy, a novel and selective therapeutic strategy, could serve as a clinical caries-prevention modality.

## 4. Materials and Methods

### 4.1. Bacterial and Phage Strains

*S. mutans UA159* and clinically isolated *S. mutans SMHBZ* served as the test organisms. Bacteria were grown in brain heart infusion (BHI) broth (Difco, Detroit, MI, USA) overnight at 37 °C under 95% air and 5% ${\mathrm{CO}}_{2}$ (*v*/*v*) conditions, or on BHI agar plates. Bacterial suspensions were adjusted to a total bacterial load of 10^6^ CFU/mL. Viable counts using serial dilutions were used to assess bacterial concentration (CFU/mL).

In this work, we used SMHBZ8 phage that was previously isolated and characterized [11]. Ben Zaken et al. [11] showed that SMHBZ8 belongs to the *Siphoviridae* family of the *Caudovirales* order with B1 morphology. Its genome accession number in the NCBI GenBank is MT430910. SMHBZ8 DNA sequencing analysis showed that it appears to be closely related to other sequenced *S. mutans* phages (M102, M102AD, and ɸAPCM01).

The phage stock was prepared as previously described [11,35]. Briefly, phages were propagated using an overnight culture of *S. mutans*, and then purified; the suspension was centrifuged at 7800 rpm for 10 min, followed by filtration through 0.22 µm filters [36]. The concentration of plaque-forming units (PFU/mL) of the phages in the stock was measured using the standard double-layered agar method [35,36]. The plates were incubated for 24 h at 37 °C under 95% air and 5% ${\mathrm{CO}}_{2}$ (*v*/*v*) conditions. The number of plaques was counted the following day and the initial concentration was calculated (PFU/mL).

### 4.2. Formulation

Hydroxypropyl cellulose was provided by local representatives of Ashland (Wilmington, DE, USA).

The material used was Klucel™ EF. The polymer was dissolved in double-distilled water (DDW) to provide a 5% solution. The DDW was gently heated on a magnetic plate to about 45 °C, the polymer was added and thoroughly dispersed, and then the temperature was lowered under constant stirring to ambience. A clear solution was obtained and used in the experiments.

Phage:polymer varnish was prepared in a 2:1 ratio by volume. Prior to each treatment, 1 mL volume of phage stock (~10^8^ PFU/mL) was mixed with 0.5 mL polymer (Klucel™ EF), and then vortexed for 2 min to a uniform solution. The varnish was air-dried at room temperature for 30 min until it solidified on the jaw/well before being subjected to growth media or bacterial infection.

### 4.3. Phage Lytic Activity against S. mutans In Vitro

Twenty microliters of polymer, phage suspension in a concentration of ~10^8^ PFU/mL, and SMHBZ8 phage:polymer mix in a 2:1 ratio (~10^8^ PFU/mL) were pipetted on the sidewalls of the wells in a vertically standing 96-well microtiter plate (cat. no. 167008, Thermo Fisher Scientific, Roskilde, Denmark).

After 30 min of solidification, 200 µL of BHI was added to each well, followed by incubation for 24 h. After 24 h, the BHI was transferred to a new 96-well plate and replaced by 200 µL of fresh BHI media. Then, 10 µL of *S. mutans SMHBZ* cultures (10^5^ CFU/mL) was added to the new plate, which was incubated overnight at 37 °C under 95% air 5% and ${\mathrm{CO}}_{2}$ (*v*/*v*) conditions. The process was repeated 4 times: 24, 48, 72, and 96 h after pipetting the suspensions.

After 24 h, the cultures were evaluated by viable counts (CFU/mL) and recorded at 37 °C in a 96-well plate reader (BioTek Synergy, BioTek, Winooski, VT, USA) at 650 nm optical density (OD)_._ Statistical significance was calculated by Student’s *t*-test (significance level: *p* < 0.05) compared to the untreated control (*n* = 5 in every tested group).

### 4.4. Polymer Toxicity Tests

Cell viability was tested as previously described [37]. For this procedure, 60,000 RAW 264.7 (ATCC TIB71) macrophage mouse cells were incubated for 24 h in a 96-well plate in Dulbecco’s modified Eagle’s medium (DMEM), high glucose, combined with 10% fetal calf serum (Biological Industries, Beit-Haemek, Israel). Then, 20 µL of each formulation (polymer only and phage:polymer) was pipetted onto the sidewalls of 5 wells in a vertically standing 96-well microtiter plate (cat. no. 1067008, Thermo Fisher Scientific, Denmark), followed by 30 min of incubation at 37 °C for solidification. Fresh media was added to each well. Cell viability was evaluated using a colorimetric XTT assay (Cell Proliferation Kit (XTT Based), Biological Industries, Beit-Haemek, Israel), as described by Scudiero et al. [38]. The XTT assay is based on the ability of metabolically active cells to reduce tetrazolium salt XTT to an orange-colored formazan compound. After 24 h of incubation at 37 °C, 50 µL of XTT, the viability indicator, was added to each well. The microtiter plate was incubated for 4 h and then monitored by optical density absorbance at 450 nm optical filter and 650 nm reference wavelength in a Vmax microplate reader (Molecular Devices Corp., San Jose, CA, USA).

### 4.5. Assessment of Phage Lytic Activity against S. mutans In Vitro in Extracted Jaw Caries Model

#### 4.5.1. Preparation of Extracted Jaws

Twenty-five hemi-mandibles were obtained from previously euthanized, healthy 10-week-old female BALB/C mice.

#### 4.5.2. Saliva Sterilization

To simulate oral cavity conditions, the samples were subjected to sterilized saliva according to Kalfas and Rnudegren [39] with a modification. Saliva was collected from healthy volunteers (authorized by the institutional ethics committee, #HMO-0706-16). Dithiothreitol (DTT) 1M (Merck KGaA, Darmstadt, Germany) was added to the saliva to reach 2.5 mM concentration. The saliva was cooled at 4 °C for 10 min and then centrifuged for 15 min. The upper fluid was discarded, and sterile double-distilled water (DDW) was added, adjusting to 25% saliva concentration. Finally, the saliva was filtered using 0.22 µm filters and frozen for reuse.

#### 4.5.3. Caries-Promoting Environment

The caries-promoting environment was established as previously described [21].

Each sample was placed in a well of a 48-well flat-bottom microtiter plate (Nunc, Thermo Fisher Scientific Inc., Waltham, MA, USA) using a sterile tweezer. The samples were subjected to 40 µL sterilized saliva for 30 min. Then, 40 µL of bacterial suspension and 400 µL of brain heart infusion (BHI) broth (Difco, Detroit, MI, USA) containing 5% sucrose were added.

#### 4.5.4. Anti-Caries Effect of Phage and Chlorhexidine

The occlusal surfaces of 20 hemi-mandible molars were subjected to 40 µL of preventive treatment before bacterial suspension and growth media were added. Four treatments were evaluated: phage suspension (~10^6^ PFU/mL), phage:polymer in 2:1 ratio (~10^6^ PFU/mL), chlorhexidine gluconate 0.2% mouthwash, and chlorhexidine digluconate 1% *w/w* dental gel (Purna Pharmaceuticals, Puurs, Belgium).

Jaws were treated and BHI was replaced every 24 h for 5 days, after which the samples were analyzed. Five jaws that were treated with polymer only served as a positive control group.

#### 4.5.5. Photographic Depiction

Each jaw was photographed using an SMZ25 stereoscope (Nikon, Tokyo, Japan) before and after the caries-promoting period. A custom-made silicone stand was prepared for each jaw, allowing a reproducible photograph to be taken in similar conditions of light and resolution.

#### 4.5.6. Micro-Computed Tomography (μCT) Evaluation of Carious Lesions

Micro-computed tomography (μCT) (μCT 40, Scanco Medical, Wangen-Bruttisellen, Switzerland) of the jaws was conducted after the experiment to quantify demineralization, as described previously [21]. Each jaw was placed in an Eppendorf tube containing 100 µL of phosphate-buffered saline (PBS) and autoclaved before the experiment.

For quantitative three-dimensional analysis of mineral tissue loss, the hemi-mandibles were examined by a desktop μCT system (μCT 40, Scanco Medical, Wangen-Bruttisellen, Switzerland), with energy of 70 kV, intensity of 114 µA, and resolution of 6 µm^3^ voxel size. Samples were placed in a cylindrical sample holder and about 300 microtomographic slices were acquired, covering the entire crown volume of each hemi-jaw. Sample scans were aligned as described by Goldman et al. [40]. Briefly, 3 points of alignment in the first molar were chosen: the mesial apical constriction, the coronal part of the mesial canal, and the distal apical constriction. After alignment, the total crown volume of the first molar was marked as a rectangle from the mesial cemento-enamel junction and adjusted accordingly for all slices in which the crown’s enamel was observed. The marked total cubic volume was then evaluated using a dedicated algorithm that gives the percentage of total volume for 6 selected density ranges (in mgHA/ccm). The caries density range was determined to be 500–1500 mgHA/ccm according to the density determined by the μCT machine.

The mean volumetric percentage of the 500–1500 mgHA/ccm range of each group, representing demineralized tissue, was compared.

#### 4.5.7. Evaluation of Viable Count

A low-speed 0.25 mm dental bur (MDT, Israel) was used to collect hard tissue surface samples from the third molars. The bur was placed in an Eppendorf tube with 100 µL PBS and sonicated for 10 min to disrupt the biofilm (Bandelin Sonopuls HD 2200, Berlin, Germany). Then, the tube was vortexed and plated on BHI agar plates (Difco, Detroit, MI, USA) for evaluation of live bacterial count (CFU/mL).

#### 4.5.8. Caries-Scoring Method for Hemi-Sectioned Molars

Caries lesions were scored using hemi-sectioned first molars, as previously described [21]. Briefly, first mandibular molars were hemi-sectioned along the mesiodistal sagittal plane using Super-Snap finishing and polishing disks (SHOFU Inc., Kyoto, Japan).

Hemi-sectioned molars were photographed using an SMZ25 stereoscope (Nikon, Tokyo, Japan). The occlusal surface of the first molar was divided into 8 surfaces, and vertical tangents for the tip of each cusp and the depth of each fissure were marked. The verticals divided the occlusal surface into 8 units. Lesions were scored as the number of occlusal surfaces that were diagnosed with caries reaching the cemento-enamel junction (CEJ).

#### 4.5.9. Statistical Analysis

The results were analyzed as the mean ± standard error of the mean of 5 samples in each experimental group. Statistical significance was calculated by Student’s *t*-test (significance level: *p* < 0.05).

### 4.6. Assessment of Phage Lytic Activity against S. mutans In Vivo in a Murine Caries Model

#### 4.6.1. Mice

Twenty-five female BALB/c mice (6–7 weeks old) were purchased from Harlan (Jerusalem, Israel). All animals were housed in ventilated cages at room temperature under a 16 h light and 8 h dark cycle. Twenty mice received 10% sucrose in distilled water and the Keyes 2000 cariogenic diet [41] ad libitum, and five mice, serving as the control group, received distilled water and a normal diet ad libitum. All animal experimental procedures were reviewed and approved by the IACUC of the Hadassah—Hebrew University Medical Center (MD-17-15315-3). Fifteen mice were subjected to bacterial infection, five mice received the high-sucrose diet only, and five mice served as the control group. Ten mice were treated every 48 h, five by oral swab with SMHBZ8 phage suspension (~10^8^ PFU/mL), and five by oral swab with chlorhexidine mouthwash 0.2%.

The time allotted for the experiment was 42 days, as in previous studies [42,43,44], and the lesions developed within 5–7 weeks.

#### 4.6.2. Bacterial Infection

*S. mutans UA159* was grown and adjusted to a total bacterial load of 10^8^ CFU/mL, similar to the in vitro experiments described above. Bacterial infection was performed every 48 h, using 0.2 mL oral gavage of bacterial suspension with 2% carboxymethyl cellulose (CMC) (Sigma-Aldrich, St. Louis, MO, USA).

#### 4.6.3. Evaluation of Bacterial Outgrowth

After 42 days, microbiological samples were collected using 20 s oral swabs from the murine mouths. The viable bacterial number was evaluated (CFU/ mL) on mitis salivarius agar for *S. mutans* and BHI agar plates for total bacterial count.

#### 4.6.4. Clinical and Radiographical Assessment of Dental Carious Lesion Development

The experiment was terminated after 42 days. Mice were anesthetized and euthanized. Hemi-mandibles were harvested. All tests were carried out as in the in vitro experiments. Hemi-mandibles were photographed using the SMZ25 stereoscope (Nikon, Tokyo, Japan) and scanned using micro-computed tomography (μCT) (μCT 40, Scanco Medical, Switzerland) for quantification of demineralization.

## 5. Conclusions

Phage therapy against *S. mutans* may serve as an efficient preventive modality against dental caries development. Optimizing the delivery would be beneficial for long-term outcomes.

## Figures and Tables

**Figure 1 antibiotics-10-01015-f001:**
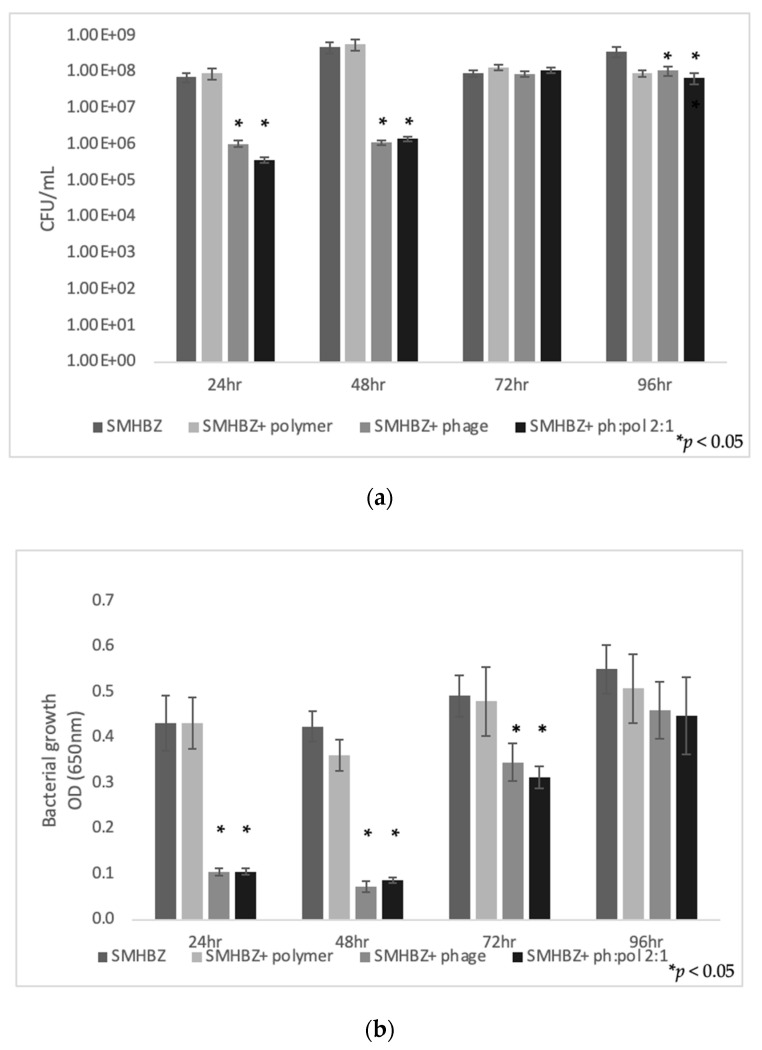
Treatment with phages and a polymer–phage formulation reduced bacterial growth of SMHBZ for 48 and 72 h compared with control groups (untreated bacteria and polymer alone), as depicted by (**a**) viable counts and (**b**) OD changes. Results are shown as mean ± SD based on five independent biological replicates.

**Figure 2 antibiotics-10-01015-f002:**
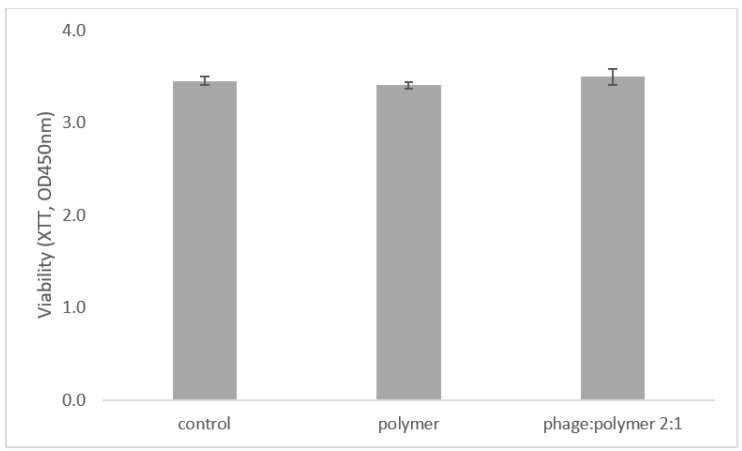
Phage:polymer formulation is not toxic to cells. XTT viability indicator shows that RAW macrophage cells were unaffected by polymer and phage:polymer in 2:1 formulation. Change in the OD_450_ of tested groups was insignificant (*p* > 0.05). Statistical significance was calculated by Student’s *t*-test (significance level: *p* < 0.05) compared to untreated control. Results are shown as mean ± SD based on five independent biological replicates.

**Figure 3 antibiotics-10-01015-f003:**
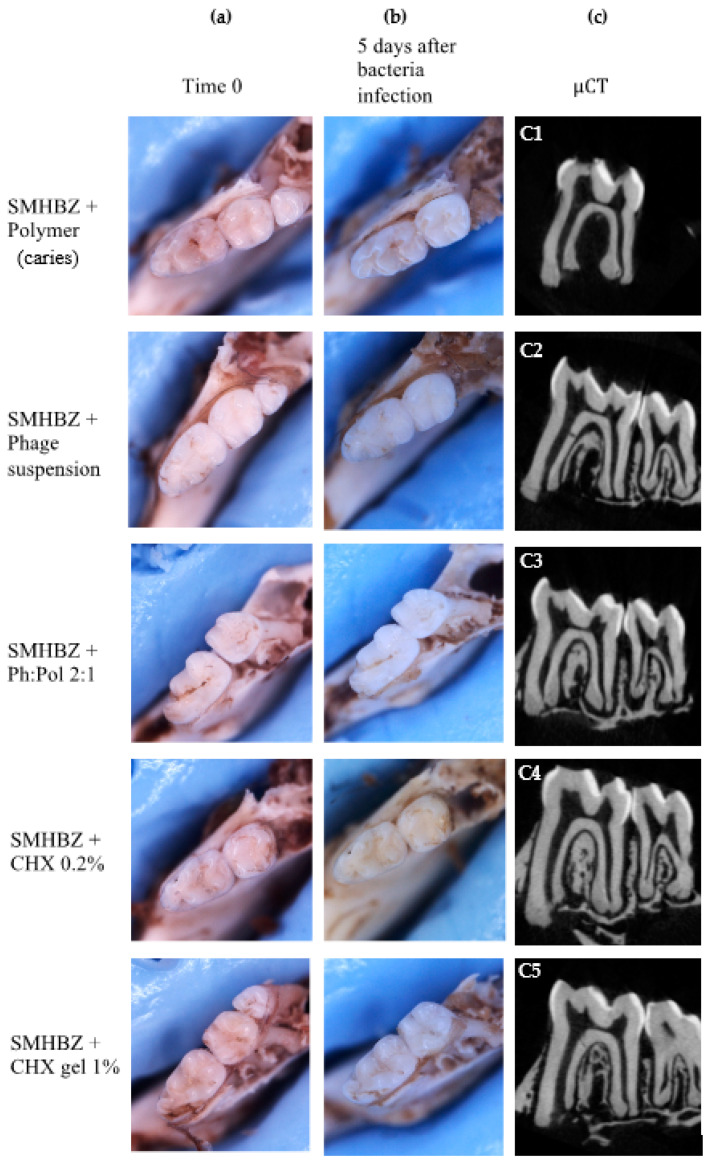
Clinical and radiographic evaluation of caries lesions after 5 days in a caries-promoting environment. Representative clinical images (**a**) before and (**b**) after caries induction using high-sucrose media and cariogenic bacteria (SMHBZ) for 5 days. (**c**) Representative radiographic image after caries induction period.

**Figure 4 antibiotics-10-01015-f004:**
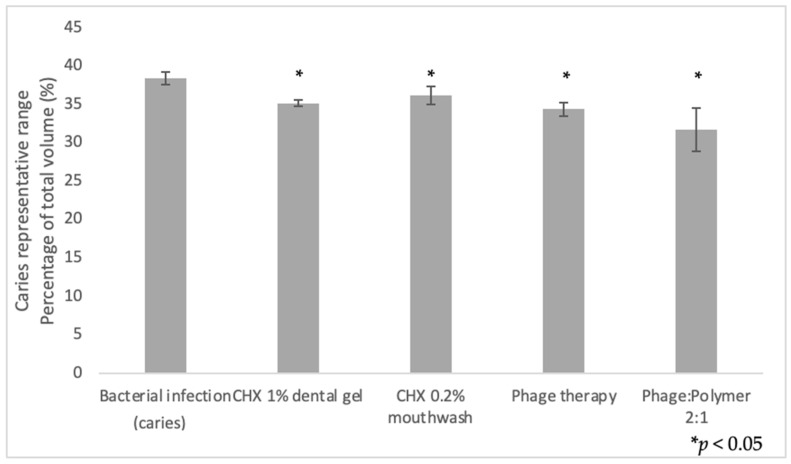
Quantification of demineralization. µCT analysis comparison of volumetric percentage in caries representative range (500–1500 mgHA/ccm) after the experimental period. 100% = total crown volume. A significant decrease (*p* < 0.05) in the caries density range was observed in all treatment groups compared to the untreated group infected with bacteria. Results are shown as mean ± SD based on five independent biological replicates.

**Figure 5 antibiotics-10-01015-f005:**
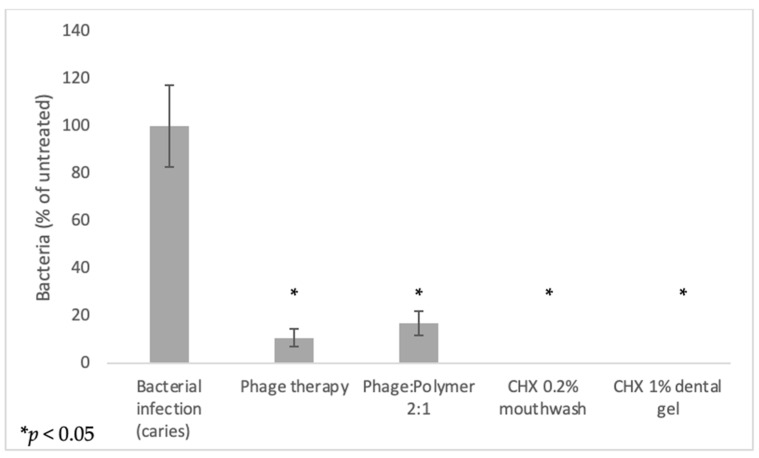
Viable bacterial counts decreased after treatment (% of untreated). Viable counts of SMHBZ bacteria (CFU/ mL) after treatment with phage suspension, phage:polymer formulation, chlorhexidine mouthwash 0.2%, or chlorhexidine dental gel 1% are described. Treatment with phage suspension and phage:polymer formulation reduced viable bacterial counts by more than 80%. Treatment with chlorhexidine-containing mouthwash and dental gel showed a bacteriocidic effect. Statistical significance was calculated by Student’s *t*-test (significance level: *p* < 0.05) compared to untreated control. Results are shown as mean ± SD based on five independent biological replicates.

**Figure 6 antibiotics-10-01015-f006:**
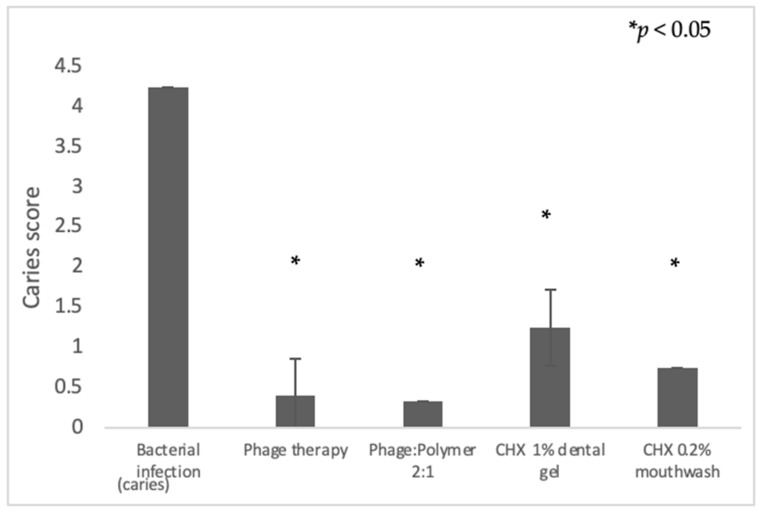
Caries scoring using hemi-sectioned molars. First mandibular molars were hemi-sectioned along the mesiodistal plane and photographed using SMZ25 stereoscope. Comparison of caries score: score increased significantly (*p* = 0.011) in jaws infected with bacteria (*n* = 5) compared to jaws treated with phages and CHX (*n* = 5). Results are shown as mean ± SD based on five independent biological replicates.

**Figure 7 antibiotics-10-01015-f007:**
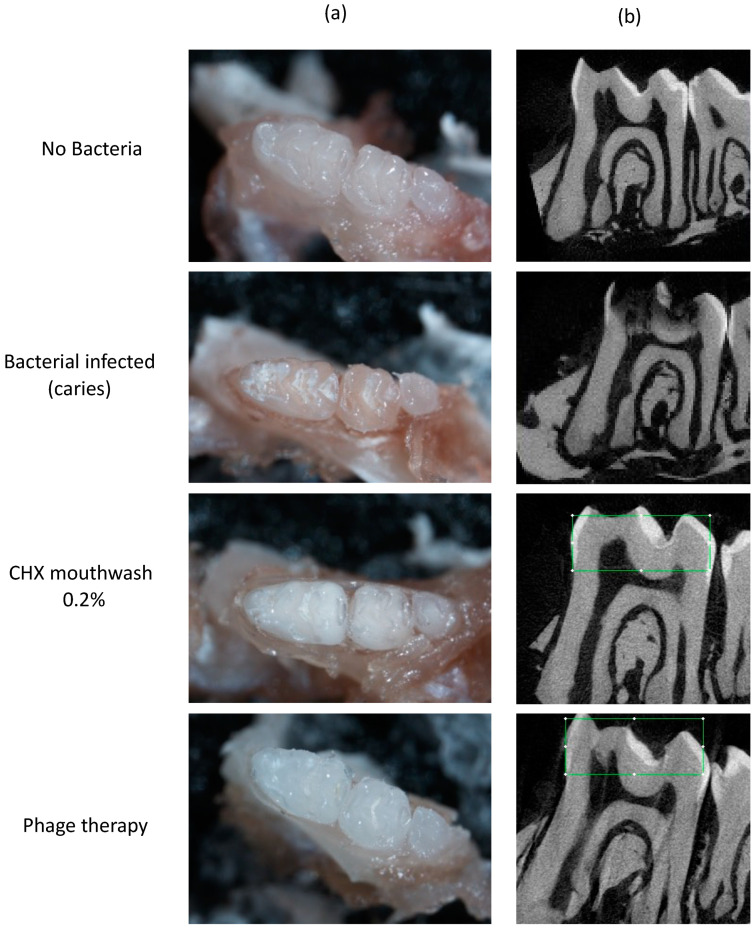
Clinical and radiographic evaluation demonstrate carious lesion development by *S. mutans* UA159 bacterial infection and high-sucrose diet. (**a**) Hemi-mandibles photographed using stereomicroscope. Representative clinical photographs depict extensive carious lesions in the group with *S. mutans UA159* bacterial infection + high-sucrose diet; no carious lesions were found in the control (no bacteria) and CHX- and phage-treated groups. (**b**) Hemi-mandibles were segmented and reconstructed to acquire 2D images by µCT.

**Figure 8 antibiotics-10-01015-f008:**
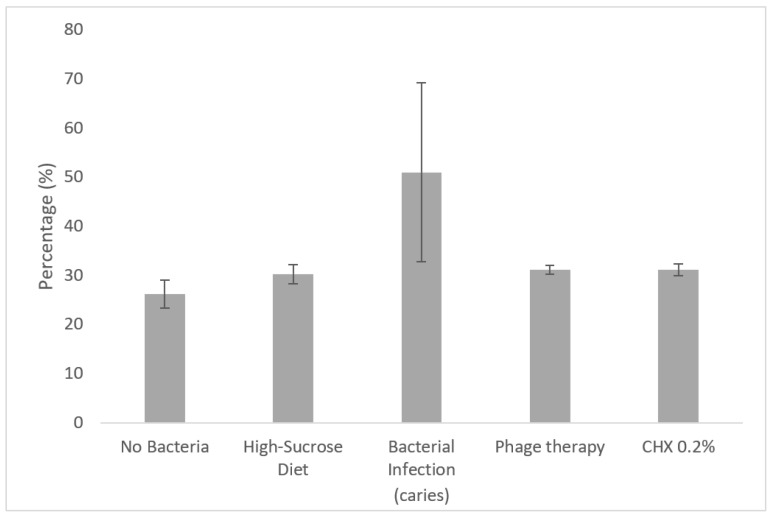
Radiographic µCT analysis shows a caries-prevention effect using phage therapy and chlorhexidine 0.2% mouthwash. µCT radiographs were analyzed using a dedicated algorithm to divide total crown volume into density ranges. Comparison of volumetric percentage of density ranges by mgHA/ccm (caries density range was determined to be 500–1500 mgHA/ccm; healthy dentin and enamel range was determined to be 1500–3000 mgHA/ccm). Mice were given a high-sucrose diet (Keyes 2000 and 10% sucrose in distilled water) and infected with *UA159* bacteria. Groups with no bacteria received a normal diet and distilled water (*n* = 5) and a high-sucrose diet (*n* = 5).

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
