# Peer review of "Phage Targeting Streptococcus mutans In Vitro and In Vivo as a Caries-Preventive Modality"

_antibiotics, 2021, doi:10.3390/antibiotics10081015_

Round 1

Reviewer 1 Report

The manuscript by Wolfoviz-Zilberman is an original article that presents preventive activity of Streptococcus mutans phage potential in caries infection which is serious problem in dentistry. The topic of this manuscript is valuable and interesting,. The structure of manuscript and the used methods are appropriate; language and style of the manuscript are rather clear. 

The suggested corrections:

  • Title: Please write Streptococcus mutans
  • Line: 51-52: Please rephrase this sentence.
  • Line 63-64: The statement :“the storage of phages causes a reduction in phage titer”  is too general.
  • Line: 73: in "the" capital letter should be change.
  • Line 85: mutans should be replace by Streptococcus mutans
  • Line 90-91: Please move this sentence to the section Materials and Methods
  • Line 92: Probably, the sentence should be incorporated in the main text of previous paragraph.
  • Some editorial work will be needed due to letter size, graphs, especially I suggest to remove the title from the top of the figure: 1, 2, 4, 5, 6c, 8. I suggest change the style of all graphs. Please check carefully if bacterial species names are written by italics.
  • Line: 306: What type of water did the authors use?

Author Response

The manuscript by Wolfoviz-Zilberman is an original article that presents preventive activity of Streptococcus mutans phage potential in caries infection which is serious problem in dentistry. The topic of this manuscript is valuable and interesting,. The structure of manuscript and the used methods are appropriate; language and style of the manuscript are rather clear. 

  • Thank you.

The suggested corrections:

Title: Please write Streptococcus mutans

  • Title has been corrected.

Line: 51-52: Please rephrase this sentence.

  • The sentence was rephrased as follows :

“Bacteriophages (phages) are viruses that infect bacteria by inserting their DNA, causing cell lysis.”

Line 63-64: The statement :“the storage of phages causes a reduction in phage titer”  is too general.

  • The paragraph was edited as follows:

One concern regarding the use of phages is their stability in solution. Phages have limited stability in solution, and there can be a significant reduction in phage titer during processing and storage [13]. Due to the fact that phages are protein structures, they are susceptible to protein denaturation factors, such as high temperature, pH, and organic solvents [13]. It was suggested that a sustained-release formulation of phages may help prolong their efficacy [13,14]. González-Menéndez et al. [15], for example, recommend encapsulating phages to maintain their stability and as a suitable method for shipment.

Line: 73: in "the" capital letter should be change.

  • Letter was changed.

Line 85: mutans should be replace by Streptococcus mutans

  • Corrected in the revised manuscript.

Line 90-91: Please move this sentence to the section Materials and Methods

  • Sentence was moved to materials and methods section.

Line 92: Probably, the sentence should be incorporated in the main text of previous paragraph.

  • The Academic editor requested to remove this sentence and add this information in figure 1 caption.

Some editorial work will be needed due to letter size, graphs, especially I suggest to remove the title from the top of the figure: 1, 2, 4, 5, 6c, 8. I suggest change the style of all graphs. Please check carefully if bacterial species names are written by italics.

  • all figures were standardized, and titles were removed. bacterial species were all checked carefully and corrected to italics where needed. Manuscript has undergone English language editing by MDPI .

Line: 306: What type of water did the authors use?

  • Double-Distilled water were used. Paragraph was corrected as follows:

“Hydroxypropyl cellulose was provided by local representatives of Ashland (Wilmington, Delaware, US).

The material used was Klucel™ EF. The polymer was dissolved in Double-Distilled water (DDW) to provide a 5-% solution. DDW was gently heated on a heating magnetic plate to about 45°C, the polymer was added and thoroughly dispersed, and then the temperature was lowered under constant stirring to ambience. A clear solution was obtained and was used in the experiments.

Phage:polymer varnish was prepared in 2:1 ratio by volume. Prior to each treatment,  1 mL volume of phage stock (~108  PFU/mL) was mixed with 0.5 mL polymer (Klucel™ EF.) and then vortex for 2 minutes to a uniform solution. The varnish was air-dried in room temperature for 30 minutes until solidification on the jaw/well before subjecting to growth media or bacterial infection.”

Reviewer 2 Report

The work of Wolfoviz-Zilberman et al is a continuous development of a phage-based product against dental caries, using a phage previously isolated and characterized by the same group. Here, they focus on in vivo murine caries models. I found the manuscript interesting, but I raise a number of concerns/observations that would like to be taken into consideration, namely:

  • Several expressions (in vitro/in vivo/ mutans) need to be italicized. Other need to be clearer. What does it mean phage:polymer 2:1 ratio? Weigh, volume, concentration…. In 2.3.2, it written: “A significant decrease was observed in a..” It may be statistically significantly, but not biologically relevant. Maximum difference observed is about 5% only.
  • Did is an important part of the work, but there are no results presented to show how reproducible and efficient is this process to trap phages. Also, what are the phage releasing kinetics? It is also not clear from the description how authors guaranty that the phages are complete dissolved in the polymer and not free during preparation. More information should be provided.
  • Some figures and legends are not clear. It is not perceptible what ph:pol 2:1 mean, why phage:polymer 1:1 was used in Figure 2 (it seems in the only case). In Figure 1, how authors can explain the fact that 0.3 OD difference represents almost 3 log ? This is surprising to me, as in most bacteria a difference of 1.0 in OD corresponds only to 1-2 logs maximum. Figures 3 e 7 are supposably caries in the control groups, but again this is not perceptible. Maybe authors can highlight it somehow.
  • Why in vivo authors did not tested phage+polymer to prevent caries? It seems only free phage was used (phage therapy).
  • Discussion needs to be modified. Nowhere authors describe what stets this phage apart from the remaining four isolated. Most importantly, there is no discussion of the difference observed between phage vs antibiotic to treat/prevent caries

Author Response

The work of Wolfoviz-Zilberman et al is a continuous development of a phage-based product against dental caries, using a phage previously isolated and characterized by the same group. Here, they focus on in vivo murine caries models. I found the manuscript interesting, but I raise a number of concerns/observations that would like to be taken into consideration, namely:

  • Thank you.

Several expressions (in vitro/in vivo/ mutans) need to be italicized. Other need to be clearer.

  • The in vitro, in vivo and bacterial species were corrected to italics where needed. Manuscript has undergone English language editing by MDPI .

What does it mean phage:polymer 2:1 ratio? Weigh, volume, concentration…

  • Information regarding varnish preparation explanation was added to “formulation” section as follows:

“Hydroxypropyl cellulose was provided by local representatives of Ashland (Wilmington, Delaware, US).

The material used was Klucel™ EF. The polymer was dissolved in Double-Distilled water (DDW) to provide a 5-% solution. DDW was gently heated on a heating magnetic plate to about 45°C, the polymer was added and thoroughly dispersed, and then the temperature was lowered under constant stirring to ambience. A clear solution was obtained and was used in the experiments.

Phage:polymer varnish was prepared in 2:1 ratio by volume. Prior to each treatment,  1 mL volume of phage stock (~108  PFU/mL) was mixed with 0.5 mL polymer (Klucel™ EF.) and then vortex for 2 minutes to a uniform solution. The varnish was air-dried in room temperature for 30 minutes until solidification on the jaw/well before subjecting to growth media or bacterial infection.”

In 2.3.2, it written: “A significant decrease was observed in a..” It may be statistically significantly, but not biologically relevant. Maximum difference observed is about 5% only.

  • All radiographic scans were standardized and there was a statistical significant in the caries representing radiographic range. The difference was 3-7% decrease in all treated groups compared to the bacterial-infected group. Clinical evaluation showed that there may be a small decrease, but it is relevant clinically- we were able to diagnose carious lesions in the bacterial infected group that were absent in the treatments’ groups.

Did is an important part of the work, but there are no results presented to show how reproducible and efficient is this process to trap phages. Also, what are the phage releasing kinetics? It is also not clear from the description how authors guaranty that the phages are complete dissolved in the polymer and not free during preparation. More information should be provided.

  • Information regarding varnish preparation explanation was added to “formulation” section as follows:

“Hydroxypropyl cellulose was provided by local representatives of Ashland (Wilmington, Delaware, US).

The material used was Klucel™ EF. The polymer was dissolved in Double-Distilled water (DDW) to provide a 5-% solution. DDW was gently heated on a heating magnetic plate to about 45°C, the polymer was added and thoroughly dispersed, and then the temperature was lowered under constant stirring to ambience. A clear solution was obtained and was used in the experiments.

Phage:polymer varnish was prepared in 2:1 ratio by volume. Prior to each treatment,  1 mL volume of phage stock (~108  PFU/mL) was mixed with 0.5 mL polymer (Klucel™ EF.) and then vortex for 2 minutes to a uniform solution. The varnish was air-dried in room temperature for 30 minutes until solidification on the jaw/well before subjecting to growth media or bacterial infection.”

The methodology included a uniform solution containing phages and the polymer was air-dried on the samples, so there were no free phages in the wells, as elaborated in the materials and methods section.  

Some figures and legends are not clear. It is not perceptible what ph:pol 2:1 mean, why phage:polymer 1:1 was used in Figure 2 (it seems in the only case).

  • Thank you for your comment. The column representing phage:polymer 1:1 was removed to avoid misunderstanding of the readers. The results were part of preliminary experiment to validate the formulation is not toxic (by examining 2 ratios).
  •  

In Figure 1, how authors can explain the fact that 0.3 OD difference represents almost 3 log ? This is surprising to me, as in most bacteria a difference of 1.0 in OD corresponds only to 1-2 logs maximum.

  • In our study, there are 3 elements that could affect the optical density of the suspension: bacterial suspension, phage suspension and the synthesized polymer. All three components may affect turbidity; the polymer is dissolved in liquid causing an increase in OD, while the phages kill the bacteria, may result in a clear solution but in anycase they reduce the overall OD.

We performed the optical density test as a microbial routine, but we believe that the more accurate results represented are bacterial viable counts.

Figures 3 e 7 are supposably caries in the control groups, but again this is not perceptible. Maybe authors can highlight it somehow.

  • Thank you for this comment. We added the word “caries” to all columns representing bacterial infected groups in the figures.

Why in vivo authors did not tested phage+polymer to prevent caries? It seems only free phage was used (phage therapy).

  • Good point. Indeed, this is a very interesting question, and it is currently under investigation in our lab but it is beyond the scope of the present study.

Recently we published an article in Applied Sciences journal, which demonstrated the significance of the in vitro model using jaws extracted from healthy mice.

As demonstrated in the article, the in vitro model is reliable and reproducible, and most importantly, can provide results similar to in vivo model.

The new article was added as a reference.

(Reference: Amit Wolfoviz-Zilberman, Yael Houri-Haddad, Nurit Beyth. A novel dental caries model replacing, refining, and reducing animal sacrifice. Applied science. 2021, 11, 7141)

Discussion needs to be modified. Nowhere authors describe what stets this phage apart from the remaining four isolated. Most importantly, there is no discussion of the difference observed between phage vs antibiotic to treat/prevent caries

  • Discussion section was modified. Paragraphs were added as follows:

Difference between this phage and the remaining isolated- we added the following sentence:

Ben-Zaken et al. [11] showed that SMHBZ8 phage was efficient in reducing S. mutans bacterial load in a dentin model, but there has been no evidence for phage therapy against S. mutans examining its ability to prevent caries or demineralization in vitro or in vivo. The current study is the first to evaluate the efficiency of phage therapy as a caries-prevention modality.”

Phage vs. antibiotics for caries prevention/treatment- we added the following paragraph:

Antibiotics are rarely used in oral diseases, as they have a limited effect on biofilms, while phages haveshown the ability to inhibit bacterial growth in biofilm [10,11,27]. Moreover, antibiotic therapy is not species-specific and can affect pathogenic as well as commensal species. Therefore, phages, which are species-specific, may be a promising alternative approach for caries treatment and prevention [6,7,27]. “
